# Measuring adverse events following hip arthroplasty surgery using administrative data without relying on ICD-codes

**Martin Magnéli**[1]*, **Maria Unbeck**[2], **Cecilia Rogmark**[3], **Olof Sköldenberg**[1], **Max Gordon**[1]

**1** Department of Clinical Sciences, Danderyd Hospital, Division of Orthopaedics, Karolinska Institutet, Stockholm, Sweden, **2** Department of Neurobiology, Care Sciences and Society, Karolinska Institutet, Stockholm, Sweden, **3** Department of Clinical Sciences, Malmö, Lund University, Lund, Sweden

\* martin.magneli@sll.se

## Abstract

### Introduction

Measure and monitor adverse events (AEs) following hip arthroplasty is challenging. The aim of this study was to create a model for measuring AEs after hip arthroplasty using administrative data, such as length of stay and readmissions, with equal or better precision than an ICD-code based model.

### Materials and methods

This study included 1 998 patients operated with an acute or elective hip arthroplasty in a national multi-centre study. We collected AEs within 90 days following surgery with retrospective record review. Additional data came from the Swedish Hip Arthroplasty Register, the Swedish National Patient Register and the Swedish National Board of Health and Welfare. We made a 2:1 split of the data into a training and a holdout set. We used the training set to train different machine learning models to predict if a patient had sustained an AE or not. After training and cross-validation we tested the best performing model on the holdout-set. We compared the results with an established ICD-code based measure for AEs.

### Results

The best performing model was a logistic regression model with four natural age splines. The variables included in the model were as follows: length of stay at the orthopaedic department, discharge to acute care, age, number of readmissions and ED visits. The sensitivity and specificity for the new model was 23 and 90% for AE within 30 days, compared with 5 and 94% for the ICD-code based model. For AEs within 90 days the sensitivity and specificity were 31% and 89% compared with 16% and 92% for the ICD-code based model.

### Conclusion

We conclude that a prediction model for AEs following hip arthroplasty surgery, relying on administrative data without ICD-codes is more accurate than a model based on ICD-codes.

**Data Availability Statement:** There are legal restrictions to upload the dataset. However, researchers interested in the dataset can contact forskning.ortopedkliniken@sll.se and will after

review and agreement to keep patient confidentiality access to the dataset. Due to the difficulty for full anonymization this restricted form of access is required. The Regional Ethical Review Board in Gothenburg: Regionala etikprövningsnämnden i Göteborg Box 401, 405 30 Göteborg; Email: registrator@etikprovning.se; Phone: +4610-475 08 00.

**Funding:** The authors received no specific funding for this work.

**Competing interests:** The authors have declared that no competing interests exist.

## Introduction

Hip arthroplasty surgery improves the quality of life for more than one million patients each year worldwide and is generally considered a safe procedure [1]. However, some patients will sustain adverse events (AEs) during or following the surgery. Rates of AEs following hip arthroplasty surgery are between 3%– 27%, depending on patient selection, measuring method and AE definition [2–4].

AEs cause both suffering for the patients and expenses for the healthcare. In a study by Culler et al. the mean cost for a hip arthroplasty surgery without an AE was \$15 600 and for a surgery with any AE was \$19 000 [5]. The cost for a surgery with $\geq$ 3AEs was \$42 900. As bundled payments and pay-per-performance are becoming more commonplace, the importance of adequate AE identification become vital from more than just a patient perspective.

Identifying and monitoring AEs is challenging. In Sweden, there have been attempts at comparing hospitals using incidence of AEs and other quality indicators. The AEs have been measured through ICD-10 codes related to readmissions in the National Patient Register (NPR). As we previously have shown the accuracy for this method is very low [6]. In addition, the reliance of ICD-codes risks introducing a coding bias in the national databases. This is well-known when diagnostic and procedural codes are connected to reimbursement [7]. In the Medicare system, self-reporting of hospital-acquired infections were biased by upcoding (mis-reporting of AEs to increase reimbursement or avoiding penalties, also known as DRG-creep) when the reporting of many infections would lead to financial penalties [8].

The aim of this study was to create a model for measuring AEs after hip arthroplasty relying on administrative data, such as length of stay and readmissions, with equal or better precision than an ICD-code based model.

## Patients and methods

### Setting and study population

This is a retrospective multi-centre cohort study on prospectively collected data from medical records and registry data [6]. The study population consisted of all patients aged $\geq$ 18 and operated with a hip arthroplasty due to osteoarthritis, hip fractures and other forms of degenerative joint disease that are registered in the Swedish Hip Arthroplasty Register (SHAR) between 2009 and 2011 (n = 21 774).

### Ethical approval

Ethical approval was provided by the Regional Ethics Committee of Gothenburg (516–13 and T732-13). Permission for data access for the reviewers was granted by the head of each respective unit. The patients did not provide an informed consent to the record review, and the need for informed consent was waived by the regional ethics committee.

### Data sources

From the SHAR we collected data on the primary procedures that were cross-linked with data from the NPR, using the Swedish personal identity numbers. From the NPR, we collected data on all admissions from the primary procedure and 90 days post-operatively. The National Board of Health and Welfare furthermore supplied aggregated data on length of stay in Swedish hospital during the study period.

We performed retrospective record review (RRR) using the Swedish version [9] of the Global Trigger Tool (GTT) [10] for all inpatient and unplanned outpatient hospital care up to 90 days after surgery. The review process is described in detail elsewhere [6, 11, 12].

## Study cohort

The study cohort consisted of 2 000 patients with both acute and elective hip arthroplasty surgery. The patients underwent surgery in one of four major regions in Sweden (Stockholm County Council, Region Västra Götaland, Region Skåne and Västerbotten County Council).

To increase the probability of selecting medical records with an AE and avoiding excess RRR on records without AEs, we used a weighted sample. 20 different selection groups for acute and elective arthroplasties were created as follows (Table 1).

1. We constructed three groups with lengths of primary stay in percentiles divided as 0–55%, 56–80% and 81–100%. The three groups were further divided based on whether there was an ICD-10 code indicating an AE in the NPR (Table 2). Overall, six groups were generated.

2. A selection was made for patients who had readmissions in the NPR. The readmission groups were divided in readmission within 2–30 days and within 31–90 days after surgery. The two groups were further divided based on whether there was an ICD-10 code indicating an AE in the NPR, generating a total of four groups.

This created 10 selection groups and we sampled according to the table, both from acute and elective patients.

## Definitions

The index admission was defined as the orthopaedic admission when the patient had hip arthroplasty surgery. If the patient was discharged directly to a geriatric or rehabilitation department, this admission was also considered a part of the index admission.

An AE in the RRR was defined as suffering, physical harm or disease as well as death related to the index admission and as a condition that was not an inevitable consequence of the patient´s disease or treatment. If an adverse event affects a patient there are in most cases also

**Table 1. Selection groups used for the weighted sample.**

With a predefined ICD-10 code indicating an AE in the NPR

|  |  | Acute | | Elective | |
| --- | --- | --- | --- | --- | --- |
|  |  | Population | Sample | Population | Sample |
| Percentiles of length of stay | 0–55% | 194 | 11 | 95 | 22 |
|  | 56–80% | 148 | 16 | 58 | 33 |
|  | 81–100% | 302 | 25 | 235 | 49 |
| Readmission | 2–30 days | 274 | 98 | 356 | 196 |
|  | 31–90 days | 199 | 98 | 204 | 195 |

Without a predefined ICD-10 code indicating an AE in the NPR

|  |  | Acute | | Elective | |
| --- | --- | --- | --- | --- | --- |
|  |  | Population | Sample | Population | Sample |
| Percentiles of length of stay | 0–55% | 2859 | 44 | 9769 | 86 |
|  | 56–80% | 1167 | 65 | 2070 | 131 |
|  | 81–100% | 766 | 97 | 1781 | 197 |
| Readmission | 2–30 days | 294 | 147 | 337 | 295 |
|  | 31–90 days | 341 | 66 | 325 | 129 |
|  | **Total** | 6544 | 667 | 15230 | 838 |

ICD-10, the 10th revision of the International Classification of Diseases.

**Table 2. Set of ICD-10 codes used in the selection of patients.**

| As main diagnosis | |
|---|---|
| All I codes | Diseases of the circulatory system |
| J819 | Pulmonary oedema |
| J13 | Pneumonia due to Streptococcus pneumoniae |
| J15 | Bacterial pneumonia, not elsewhere classified |
| J18 | Pneumonia, organism unspecified |
| R33 | Retention of urine |
| As main or secondary diagnosis | |
| I803 | Phlebitis and thrombophlebitis of lower extremities, unspecified |
| I269 | Pulmonary embolism without mention of acute cor pulmonale |
| L899 | Decubitus ulcer and pressure area, unspecified |
| M243 | Pathological dislocation and subluxation of joint, not elsewhere classified |
| M244 | Recurrent dislocation and subluxation of joint |
| S730 | Dislocation, sprain and strain of joint and ligaments of hip |
| T810 | Haemorrhage and haematoma complicating a procedure, not elsewhere classified |
| T813 | Disruption of operation wound, not elsewhere classified |
| T814 | Infection following a procedure, not elsewhere classified |
| T840 | Mechanical complication of internal joint prosthesis |
| T845 | Infection and inflammatory reaction due to internal joint prosthesis |
| T933 | Sequelae of dislocation, sprain and strain of lower limb |

ICD-10, the 10th revision of the International Classification of Diseases.

suffering, i.e. something subjective unpleasant, involved. If, for example, a patient is affected by a deep wound infection with a long hospital stay and reoperations there is inevitably suffering involved in connection to this along with physical harm. Suffering is closely connected to the physical harm or disease, and death part of the AE definition used in our study. An inevitable consequence means that the adverse event is associated with healthcare-related omissions or commissions rather than an underlying disease or injury of the patient.

The outcome was at least one AE of any type or severity.

## Data set

Two patients were excluded, one did not have an available medical record and the other did not have hip arthroplasty surgery and was presumed faulty registered in the SHAR. The final study cohort consisted of 1 998 patients. Of these patients, 1 171 had at least one AE (the high proportion of AEs was due to the selection of the cohort which targeted groups with high probability for AEs, see above section). Predictor variables included gender, age, length of primary stay (LOS) both for the orthopaedic admission and the index admission, acute or elective procedure, type of hospital for the primary surgery (university, central county council, county council or private), number of readmissions, number of emergency department (ED) visits and if the patient was discharged to an acute care ward (surgical, internal medicine, cardiology, infection or intensive care). We cross-linked the patient data with aggregated data matching the patient's age, gender, acute or elective care, year of surgery and type of hospital with the aggregated data. The aggregated data included the 50th, 75th, 90th and 95th percentiles, mean and standard deviation of LOS of that patient and hospital category. This resulted in that each patient had in addition to their own LOS, aggregated data for their patient characteristics. We used the mean of these 50th percentiles to calculate the LOS trends in Sweden during the study period.

**Table 3. The set of ICD-10 codes for the defining an adverse event by the instrument.**

| As main diagnosis | |
| --- | --- |
| All I codes | Diseases of the circulatory system |
| J819 | Pulmonary oedema |
| J13 | Pneumonia due to Streptococcus pneumoniae |
| J15 | Bacterial pneumonia, not elsewhere classified |
| J18 | Pneumonia, organism unspecified |
| R33 | Retention of urine |
| As main or secondary diagnosis | |
| L899 | Decubitus ulcer and pressure area, unspecified |
| S730 | Dislocation, sprain and strain of joint and ligaments of hip |
| T810 | Haemorrhage and haematoma complicating a procedure, not elsewhere classified |
| T813 | Disruption of operation wound, not elsewhere classified |
| T814 | Infection following a procedure, not elsewhere classified |
| T840 | Mechanical complication of internal joint prosthesis |
| T845 | Infection and inflammatory reaction due to internal joint prosthesis |
| T933 | Sequelae of dislocation, sprain and strain of lower limb |

ICD-10, the 10th revision of the International Classification of Diseases.

## Reference model based on ICD-codes (code-model)

The model used by both the SHAR and the National Board of Health and Welfare is based on a set of ICD-10 codes (Table 3). According to the model, a patient has sustained an AE if any of the codes is present in the NPR during readmissions. This model was used as our reference model.

## Model development

The full technical description of the model development is available in the S1 Appendix. We made a 2/1 split of the data into a training and holdout dataset. The training data was used to train a set of machine learning algorithms (random forests, logistic regression with and without natural splines, support vector machines and neural networks with three different structures). We used 10-folds cross validation to evaluate model accuracy, fine tune hyper parameters and control for over-fitting. During training, we started with all variables included in the model and did a stepwise removal of variables. We also split the dataset into acute and elective surgery and trained two separate models for each set. The best performing and fastest model was s logistic regression model with four natural age splines. Fig 1 shows a flow chart of the model development.

## Validation of models

In the final test, we used the whole training set to train the model and the trained model was used for prediction for the holdout set. We made a prediction for each selection group. The sensitivity and specificity in the groups were multiplied by the group proportion (group size in population/total population) and summed, this yielded the adjusted sensitivity and specificity. The final model was evaluated against the holdout set. For the code-based model, the sensitivity, specificity and Youden's index for the code-based model on the holdout data was calculated using the same method as on the training data.

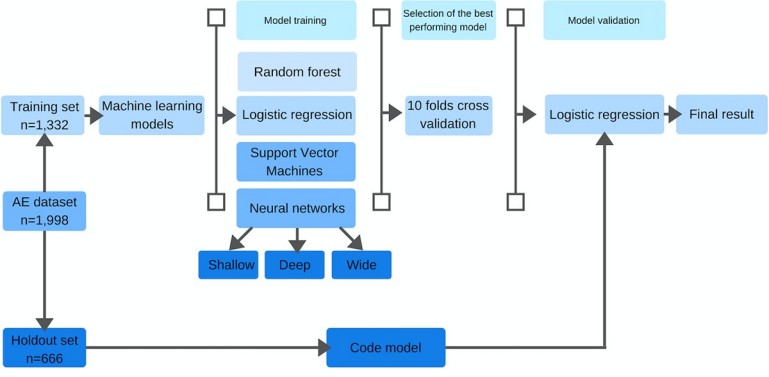

**Fig 1.**

## Performance metrics

We compared the models with the code-model by measuring sensitivity, specificity and Youden Index (sensitivity + specificity– 1) [13]. For intermodal comparisons, we relied on the area under the receiver operator characteristic curve (AUC). The receiver operator characteristic (ROC) is created by plotting a curve of the different classification thresholds on the true positive and false positive rates. The AUC is the two-dimensional area under this curve. This curve could not be calculated for the code-model because the result from this model are dichotomous and does not contain any thresholds, we therefore used AUC during the model training and Youden Index for the validation of the final model.

## Software and packages

We used R 3.5.1 for all statistics. We used the stats package for logistic regressions and the rms package (v.5.1–2) and the contrast function for calculating odds ratio and 95% confidence interval (CI) for age and LOS. The graphs were created using ggplot2 (v. 3.0.0). The packages used for the different models are available in the S1 Appendix.

## Results

One third of the participants in the study cohort were treated due to hip fractures (acute group) and two thirds due to degenerative joint disease (elective group). The acute patient group consisted of more women, with a higher median age and longer LOS (orthopaedic admission with following rehabilitation admission) than the elective patients (Table 4). There were no large differences in median age, AE proportion, gender, median LOS or acute or elective operation between the training and holdout set (Table 5).

### Training results

The performance difference between the models was negligible. The AUC from the training were similar for all the models. Most models had a slightly higher AUC when we included ICD-codes than without codes (Fig 2). For the three different configurations of neural networks, no configuration was superior, and all neural networks had inferior performance compared to the traditional machine learning models (Fig 3).

### Best performing model results

The best performing model, logistic regression with four natural age splines (henceforth: top model) had higher sensitivity, specificity and Youden's Index for both 30 and 90 days when

**Table 4. Demographics.**

|  | All patients n = 1 998 | Acute group n = 667 | Elective group n = 1 331 |
|---|---|---|---|
| Female, n (%) | 1 250 (62.6) | 444 (66.6) | 806 (60.6) |
| Male, n (%) | 748 (37.4) | 223 (33.4) | 525 (39.4) |
| Age, median (IQR) | 77 (68–84) | 84 (79–89) | 73 (64–80) |
| LOS, median (IQR) | 7 (4–13) | 14 (9–20) | 5 (4–8) |
| Type of hospital, n (%) |  |  |  |
| University | 630 (31.5) | 295 (44.2) | 335 (25.2) |
| Central county council | 556 (27.8) | 180 (27.0) | 376 (28.2) |
| County council | 531 (26.6) | 109 (16.3) | 422 (31.7) |
| Private | 281 (14.1) | 83 (12.4) | 198 (14.9) |

LOS, Length of stay; IQR, Interquartile range.

Note: Weighted sample, the mean values are not representable for average Swedish orthopedic care.

tested on the acute, elective and all patients (Table 6). We started with all variables in the model and then removed them one by one. The best performance was observed using length of stay at the orthopaedic department, discharge to acute care, age, number of readmissions and ED visits.

The precision was higher for all patients than for both acute and elective patients. We analysed the relative importance of the variables in the top model and readmission and number of ED visits were the two most important variables (Table 7). We weighted the top model to include more negative cases to match the specificity of the code-based model. This way the precision could be compared using the sensitivity. We also tried the top model with ICD-codes but this weakened the results instead of improving them.

## Other analyses

An increased LOS was associated with an increased risk of having a registered AE (Fig 4). Also increased age was associated with an increased risk of having a registered AE (Fig 5).

## Discussion

### Key results

Our alternative model without any ICD-codes outperformed the reference code-based model. It was able to attain the same specificity while having 2–3 times the sensitivity of the code-

**Table 5. Demographics for the training and holdout set.**

|  | Training set n = 1 332 | Holdout set n = 666 |
|---|---|---|
| Age, median (IQR) | 77 (67–84) | 78 (69–85) |
| AEs, n (%) | 781 (58.6) | 390 (58.6) |
| Female, n (%) | 840 (63.1) | 410 (61.6) |
| Male, n (%) | 492 (36.9) | 256 (38.4) |
| LOS, median (IQR) | 6 (4–9) | 6 (4–8) |
| Acute | 438 (32.9) | 229 (34.4) |
| Elective | 894 (67.1) | 437 (65.6) |

LOS, Length of stay; IQR, Interquartile range.

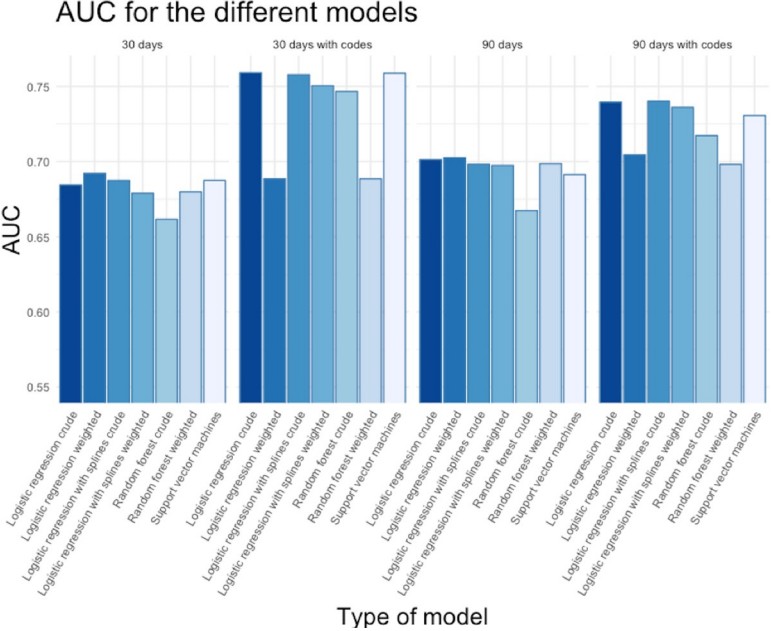

**Fig 2.**

based model. The strongest indicator for the occurrence of an AE were the number of readmissions and ED visits.

We found that the risk of having a registered AE occur increases with longer LOS and increased age. LOS is naturally dependent on how the healthcare is organized. In Sweden, the median LOS for hip fracture patients in the orthopaedic ward is 7 days and after this is transferred to either a geriatric ward, nursing home or home with or without home healthcare or social care. We used LOS for the orthopaedic stay and not the combined LOS of the orthopaedic and geriatric stay because this improved the model accuracy. This is logical considering that most AE occurred during the orthopaedic stay. In other healthcare systems the patient may stay shorter in the orthopaedic ward and is discharged to step-down facilities or longer if the rehabilitation is done in the orthopaedic ward. In these systems, the occurrence of an AE may not affect LOS as much as in the Swedish system.

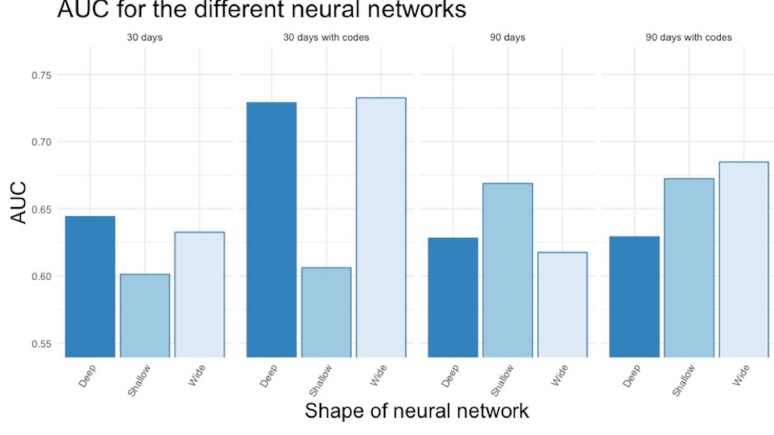

**Fig 3.**

**Table 6. Results comparing the reference code model with the top performing logistic regression model.**

| | 30 days | | | 90 days | | |
|---|---|---|---|---|---|---|
| | Sensitivity | Specificity | Youden's index | Sensitivity | Specificity | Youden's index |
| *All patients* | | | | | | |
| Code model | 0.054 | 0.942 | -0.005 | 0.164 | 0.915 | 0.079 |
| Top model | 0.230 | 0.906 | 0.136 | 0.314 | 0.894 | 0.208 |
| *Acute patients* | | | | | | |
| Code model | 0.107 | 0.865 | -0.028 | 0.277 | 0.767 | 0.044 |
| Top model | 0.319 | 0.779 | 0.098 | 0.464 | 0.758 | 0.221 |
| *Elective patients* | | | | | | |
| Code model | 0.035 | 0.976 | 0.010 | 0.050 | 0.962 | 0.012 |
| Top model | 0.073 | 0.937 | 0.010 | 0.090 | 0.937 | 0.028 |

There was barely any performance difference in the tested models. We suspect that this is due to that the amount of information in the administrative data is limited and can be adequately captured using standard statistical models. This was supported by the fact that the most complex models such as neural networks performed even worse than the simpler models.

Depending on the purpose of the top model, it can be adjusted to gain a higher sensitivity or specificity by changing the cut point or the case weights. A model for economic purposes might be adjusted to elevate the specificity to ensure a low true negative rate.

## Strengths and limitations

The main strength in this study is the use of a large multi-centre data set with high quality data, and probably most important that all the AEs were validated with RRR. RRR with GTT is the method that will detect most AEs [14–16], but still it is limited to the information recorded in the records. Also, RRR with GTT is both time and resource consuming. The variables used in the model are robust and easy to measure. An interesting finding is that the model with less variables performed better than when all variables were included. The variety of different AEs is wide; however, they seem to only affect only a few variables found in administrative data. We interpret this as a sign that this dataset is not complex, and this is an explanation why the more advanced machine learning models did not outperform basic statistic models. If reimbursement to hospitals is based on short LOS and few readmissions instead of ICD-codes, this

**Table 7. Importance of the variables in the logistic regression model.**

| | Estimate | Standard Error | z value | Pr(>\|z\|) |
|---|---|---|---|---|
| Intercept | -0.649 | 0.940 | -0.690 | 0.490 |
| LOS | 0.058 | 0.022 | 2.629 | 0.009 |
| Readmissions | 0.567 | 0.077 | 7.361 | >0.001 |
| ED visits | 0.685 | 0.117 | 5.846 | >0.001 |
| Discharge to acute care | 1.854 | 0.677 | 2.738 | 0.006 |
| Age spline 1 | -1.102 | 0.857 | -1.286 | 0.198 |
| Age spline 2 | -0.427 | 0.616 | -0.694 | 0.488 |
| Age spline 3 | -0.635 | 1.951 | -0.325 | 0.745 |
| Age spline 4 | 0.496 | 0.916 | 0.542 | 0.588 |

LOS, Length of stay; ED, emergency department.

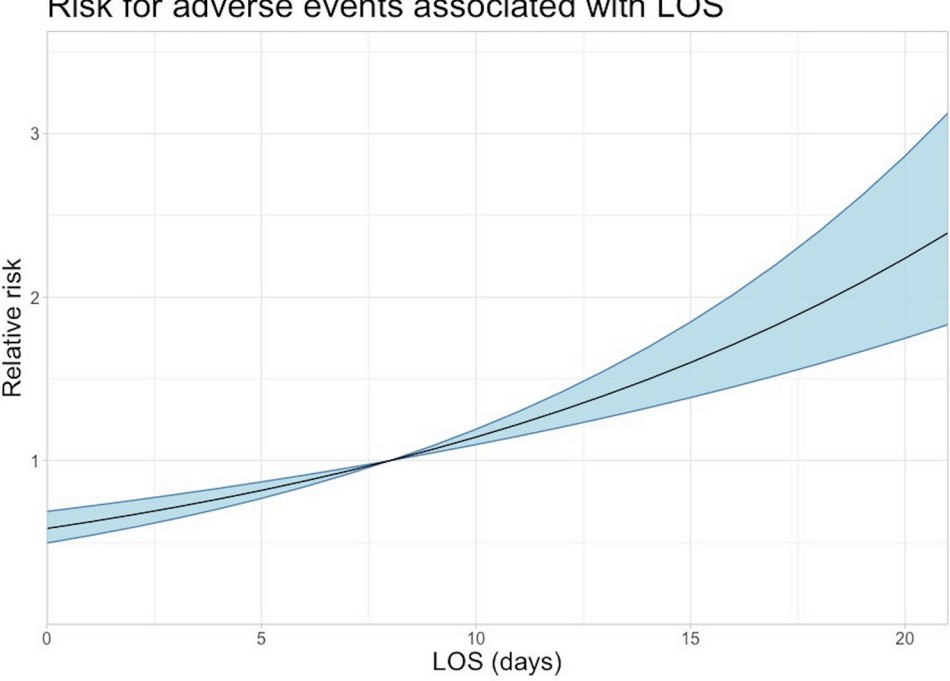

**Fig 4.**

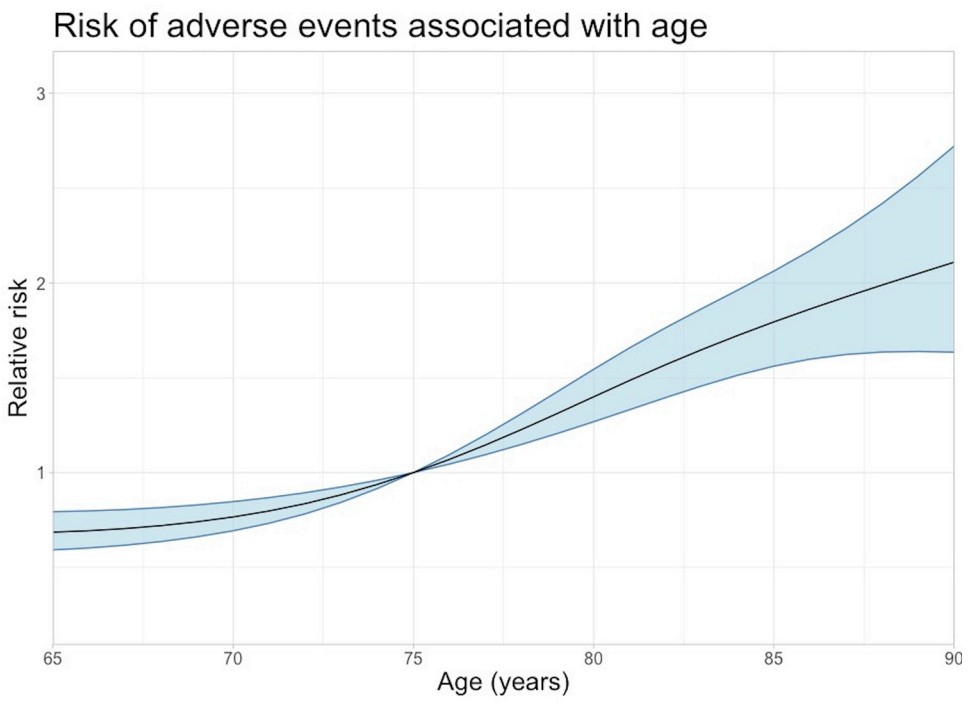

**Fig 5.**

might stimulate hospitals to improve these variables which will unlock resources for other patients. Compared to upcoding the side effects from weighting in LOS is much more positive.

This study explores the use of many different machine learning methods. The use of neural networks was not more accurate than the other methods. Neural networks have become very popular in recent years, especially the use of convolutional neural networks for image classification [17]. The result of this study can be a healthy reminder that this method is maybe not always the best choice for all type of prediction and it could be worthwhile to try different methods.

The use of a weighted sample has the advantage of recording many AEs with minimum record review and it will generate a dataset that is more balanced concerning the outcome. However, the results have to be adjusted to represent the results in the study population.

Notably, also legislation on confidentiality has to be considered when designing models to monitor AE. Our study was delayed due to the bureaucracy to require all records in this national multi-centre study. The lack of a unanimous definition of what should be considered as AE hinders comparisons between studies and countries.

This study only includes limited patient demographic data (age and gender) and lacks some important demographics as comorbidities, smoking status and BMI, which is often found in the medical record, but sometimes in the administrative data.

Even though the accuracy of the top model is higher than the code model, it is limited and there is room for improvement. To improve it there is probably a need to add data beyond the NPR. The improvement might come from adding data that might correspond to certain individual AEs. One possibility would be to add data from the Swedish Prescribed Drug Registry that collects data on all prescribed drugs that are delivered to Swedish patients. If a patient is prescribed antibiotics or high-dose anticoagulants following surgery, this could be a proxy for infection or thrombosis that could be included and improve the model.

### Interpretation

Risk adjusted prolonged length of stay (RAPLOS) as a measure for AEs following colon resection, coronary artery bypass graft and hip arthroplasty have been studied by Fry et al. [18]. The authors concluded that RAPLOS was a better measure for AEs than codes. However Lyman et al. [19] studied RAPLOS as a measure for AEs after following elective hip and knee arthroplasty surgery and concluded that RAPLOS was not superior to a measure based on ICD-codes. This study did not rely on RRR for measuring AEs and our model uses more variables which makes comparison difficult.

### Generalizability

The administrative data used in the model is universal and easily available through hospital administration systems, which would enable use of the model worldwide. AEs causes prolonged LOS [20–23] and readmissions are also correlated with AEs [24] and unplanned readmission can be used as a proxy for AEs [25]. Based on this knowledge a model based on LOS and readmissions is probably applicable to other types of surgery and developing such models for other types of surgery could probably be done with less patients than in this study.

### Conclusion

We conclude that a prediction model for AEs following hip arthroplasty surgery based on administrative data without ICD-codes is more accurate than a model based on ICD-codes. In addition to the accuracy, variables such as LOS, readmissions, gender and age are robust and

objective. Therefore, they are not prone to be biased in a way that ICD-codes can be. We consider that this less is more model is superior to ICD-code based models.

## Patient involvement

This is a register and record-based retrospective study with no patient involvement.

## Supporting information

**S1 Appendix.**
(DOCX)

## Acknowledgments

The authors thank Marie Ax, Susanne Hansson, Ammar Jobory, Zara Hedlund, Mirta Stupin, Tim Hansson, Lovisa Hult-Ericson and Christina Jansson for valuable help in carrying out the study. We would also like to thank all department managers for access to the medical records and Per Nydert for help with the study database.

## Author Contributions

**Conceptualization:** Cecilia Rogmark, Olof Sköldenberg, Max Gordon.

**Formal analysis:** Martin Magnéli.

**Methodology:** Maria Unbeck.

**Writing – original draft:** Martin Magnéli.

**Writing – review & editing:** Martin Magnéli, Maria Unbeck, Cecilia Rogmark, Olof Sköldenberg, Max Gordon.

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
