## [Decision Letter · Decision Letter 0]

20 Aug 2020

PONE-D-20-09967

Developing and validating a model for measuring adverse events following hip arthroplasty surgery using administrative data without ICD-codes

PLOS ONE

Dear Dr. Magneli,

Thank you for submitting your manuscript to PLOS ONE. After careful consideration, we feel that it has merit but does not fully meet PLOS ONE’s publication criteria as it currently stands. Therefore, we invite you to submit a revised version of the manuscript that addresses the points raised during the review process.

Your manuscript has been reviewed by two experts in the field, who request more information and clarification mostly about specifics on modelling.

We look forward to receiving your revised manuscript.

Kind regards,

Susan Hepp

Academic Editor

PLOS ONE

Journal Requirements:

2. Please provide additional details regarding participant consent. You state that "the patients did not provide an informed consent to the record review, which was accepted by the local ethics committee". Please clarify if the need for consent was waived by the ethics committee. Alternatively, please discuss whether all data were fully anonymized before you accessed them.

3.We suggest you thoroughly copyedit your manuscript for language usage, spelling, and grammar. If you do not know anyone who can help you do this, you may wish to consider employing a professional scientific editing service.  

4.We note that you have indicated that data from this study are available upon request. PLOS only allows data to be available upon request if there are legal or ethical restrictions on sharing data publicly. For information on unacceptable data access restrictions, please see http://journals.plos.org/plosone/s/data-availability#loc-unacceptable-data-access-restrictions.

5. Your ethics statement must appear in the Methods section of your manuscript. If your ethics statement is written in any section besides the Methods, please move it to the Methods section and delete it from any other section. Please also ensure that your ethics statement is included in your manuscript, as the ethics section of your online submission will not be published alongside your manuscript.

Additional Editor Comments (if provided):

1. Please provide additional details regarding participant consent. You state that "the patients did not provide an informed consent to the record review, which was accepted by the local ethics committee". Please clarify if the need for consent was waived by the ethics committee. Alternatively, please discuss whether all data were fully anonymized before you accessed them.

2. At this time, please thoroughly copyedit your manuscript for language usage, spelling, and grammar. If you do not know anyone who can help you do this, you may wish to consider employing a professional scientific editing service.   Whilst you may use any professional scientific editing service of your choice, PLOS has partnered with both American Journal Experts (AJE) and Editage to provide discounted services to PLOS authors. Both organizations have experience helping authors meet PLOS guidelines and can provide language editing, translation, manuscript formatting, and figure formatting to ensure your manuscript meets our submission guidelines. To take advantage of our partnership with AJE, visit the AJE website (http://learn.aje.com/plos/) for a 15% discount off AJE services. To take advantage of our partnership with Editage, visit the Editage website (www.editage.com) and enter referral code PLOSEDIT for a 15% discount off Editage services. If the PLOS editorial team finds any language issues in text that either AJE or Editage has edited, the service provider will re-edit the text for free. 

Reviewers' comments:

Reviewer's Responses to Questions

**Comments to the Author**

1. Is the manuscript technically sound, and do the data support the conclusions?

Reviewer #1: Yes

Reviewer #2: Yes

2. Has the statistical analysis been performed appropriately and rigorously? 

Reviewer #1: Yes

Reviewer #2: I Don't Know

3. Have the authors made all data underlying the findings in their manuscript fully available?

Reviewer #1: No

Reviewer #2: Yes

4. Is the manuscript presented in an intelligible fashion and written in standard English?

Reviewer #1: Yes

Reviewer #2: Yes

5. Review Comments to the Author

Reviewer #1: Review of the manuscript for PlosONE by Magneli et al developing and validating a model for measuring adverse events following hip arthroplasty surgery using administrative data without ICD-codes. The research groups hypothesized that they could use administrative data independent of ICD-codes to create a new model with equal or better ability to measure adverse events after hip arthroplasty. They concluded that a prediction model for AEs following hip arthroplasty surgery based on administrative data without ICD-codes is more accurate than a model based on ICD-codes.

This is a well written paper with a clear message. The research group did a tremendous job by combining the datasets of the SHAR and the National Patient Register. Additionally using aggregated data from the National Board of Health and Welfare and performing retrospective record review using the Global Trigger Tool. They performed sound statistics with a large variety of testing models which increased strength of this research.

However, there are some minor points of attention.

What about missing data? Wat are the proportions of missing data for each variable? Does the medical files contain close to all variables for each patient? Is the missing data missing at random?

Several different models were tested, with rather similar outcomes on sensitivity and specificity. As you speculate in the discussion this might be caused by the limited number of variables include. Could the model(s) be improved by adding any additional and often available variables? Which variables do you expect to improve the model most?

Length of stay at the orthopaedic department is an important variable in your best model, especially in the Swedish situation so you describe in your discussion. Can you elaborate on this ‘length of stay’ variable. Does it possibly contains several unmeasured factors bundles as 'length of stay' at the orthopaedic department? And therefore, how generalizable is your model to other countries?

Do you have any suggestions as how to use the results of your study to improve orthopaedic care? Can you elaborate on that?

I am pleased with the Figure 1 showing clearly the data and analysis steps. I would suggest to use the same y-axis scale in figure 4 and 5 and clearly state in the title that it concerns Risk of adverse events associated with age/LOS. In my opinion figure 6 is not so informative and these results could be mentioned in the text of the Results section (an/or figure in an appendix).

Reviewer #2: I would like to thank the editorial office for giving me the chance to review

"Developing and validating a model for measuring adverse events following hip arthroplasty surgery using administrative data without ICD-codes" by Magnéli et al.

I interpret this study as an exploratory (diagnostic) study on retrospective data.

The overall idea with the study is of utmost importance, and I would recommend acceptance of the manuscript.

I do however have comments to the manuscript, which I believe would improve the readability.

General comments:

It is a very technical paper, and it is hard to follow the red thread from introduction to conclusion. Many models are evaluated on the same data set, and commented on throughout the manuscript, but this study is not on the performance of different AI models, but whether administrative data captures AE better than ICD codes (to my understanding, or do I miss something?). But this also needs to reflect in the title, is it model development or adm. data vs ICD testing or perhaps both you look into.

I would suggest revision of the entire manuscript with focus on a single chosen model (the best performing) and simply explain, perhaps in an appendix, that the model was chosen based on sensitivity analysis of many models, and explain the models in the appendix. And then focus the manuscript on this models performance against the gold standard of an ICD code model. Which is what I believe is the authors purpose??

The statistical modeling is complex, and beyond the abilities of this reviewer to evaluate in specifics. A notion of the experience of the author group in DL modeling would thus be nice. And also a passage on how easy the model can be constructed for use in other healthcare systems?

It is not clear why the individual administrative data was selected for modelling? Where they simply available or did the authors reflect a priori on this. Needs to be adressed.

I would like if the authors state their opinion on what is next - is this model only applicable to Swedish system or should it be validated locally before application. And what is the clinical perspective of the model, is it only for use in administrative settings - such as comparison between centers, or can clinician locally or nationally benefit from this model?

A discussion of whether the lower specificity of the new model compared to the ICD model is actually beneficial in the context of use by administrators to oversee surgeons.

Do the authors plan to perform a validation of the model in a low prevalent AE group as well?

Specific comments:

line 36: must be 30 days instead of 90 days?

Line 69: I do not like the word "hypothesize" in a retrospective, exploratory study

Line 94-106 it is very difficult to comprehend how, and why the groups were constructed, and to what extend this influence the outcome. Why did you want to increase the probability of AE in the sample? Other causes than just for ease of journal review? Also later on discuss how the prevalence of AE in the selected sample influence the accuracy of your models. It is a highly selected dataset which could influence the conclusion. I do not understand Aggregated LOS data? So not individual LOS data?? You discuss later in text but it is not easily understood.

line 98 add "on a dedicated sample of patients" after surgery and before the period sign.

line 112: "suffering" needs to be defined. In general this is a very loose description of AE, which makes it very difficult for others to replicate your study if desired. Needs to be more clearly stated.

line 113-114. - please define "inevitable consequences". See line 112 above.

line 115-117 why is the definition of index admission placed under AE definition subheading?

line 123 & line 130 I do not understand the difference in the context of this study. Why is this necessary?

Line 134 why is LOS trends valuable information in this study?

Line 161-162 rephrase to a more scientific description than "we tried to"....

Line 204-205 Does this not mean that we should use BOTH administrative data AND ICD codes in future models, instead of only either? Needs to be discussed.

Line 220 Only 2/3 were elective, non-fracture patients. Needs to address the consequences - sensitivity analysis. : line 220 "The precision was higher for all patients than for both acute and elective patients." what does this mean for your model and for external validity?

Line 226-227: Needs to be discussed in the discussion section as to potential pitfalls of the model.

Line 248 "We found that the risk for sustaining an AE increased with longer LOS" - I would not use the word "risk of sustaining". I would use risk for having an registered AE occur increases with longer LOS. Sustaining an AE during admission will lead to longer LOS, LOS does not increase the risk of sustaining an AE during admission in your data set. also since line 253 states: This is logical considering that most AE occurred during the orthopaedic stay.

line 268 "the AEs were collected with RRR." Could AEs occur without being registered. I would use term "validated" instead of collected, and add sentence about missing potential AEs.

line 262 and line 269 "An interesting finding is that the model with less variables performed better than when all variables were included. ". You place this under strength of your study, but you lack any further discussion into this. This goes as well for the complexity in the model, which gives a worse performance. These two findings has also been found by others in similar AI fields (Lauritsen et al. 2020 Artificial Intelligence in Medicine.). The manuscript would benefit with a discussion of this.

6. PLOS authors have the option to publish the peer review history of their article (what does this mean?). If published, this will include your full peer review and any attached files.

Reviewer #1: No

Reviewer #2: **Yes: **Jeppe Lange

---

## [Author Response · Author response to Decision Letter 0]

21 Sep 2020

Editors comments

Comment: Please provide additional details regarding participant consent. You state that "the patients did not provide an informed consent to the record review, which was accepted by the local ethics committee". Please clarify if the need for consent was waived by the ethics committee. Alternatively, please discuss whether all data were fully anonymized before you accessed them.

Answer: The need for written consent was waived by the ethics committee. We have added this in the manuscript.

C: We note that you have indicated that data from this study are available upon request. PLOS only allows data to be available upon request if there are legal or ethical restrictions on sharing data publicly.

A: There are legal restrictions to upload the dataset. However, researchers interested in the dataset can contact forskning.ortopedkliniken@sll.se and will after review and agreement to keep patient confidentiality access to the dataset. Due to the difficulty for full anonymization this restricted form of access is required. The Regional Ethical Review Board in Gothenburg: Regionala etikprövningsnämnden i Göteborg Box 401, 405 30 Göteborg; Email: registrator@etikprovning.se; Phone: +4610-475 08 00.

C: Your ethics statement must appear in the Methods section of your manuscript. If your ethics statement is written in any section besides the Methods, please move it to the Methods section and delete it from any other section. Please also ensure that your ethics statement is included in your manuscript, as the ethics section of your online submission will not be published alongside your manuscript.

A: We have moved the ethics statement to the methods section.

Reviewer 1

1 Question: What about missing data? Wat are the proportions of missing data for each variable? Does the medical files contain close to all variables for each patient? Is the missing data missing at random?

Answer: Thank you for this important question, we had no missing data for the variables used in the mode (LOS, discharge to acute care, age, number of readmissions and ED visits) since these variables emanated from the national patient registry, which is very complete. All these variables were also available in the medical records. In the aggregated data there were a few missing datapoints (n=10). The explanation is not the completeness of the registry data, but some younger patients were the only patient in that age/sex/hospital type group during that specific year. So, these patients can be considered outliers in age and not completely missing at random. However, since the missing data patients was only a fraction (0.5%) we decided to exclude them in the initial model training and testing. We added these patients when we discovered that these aggregated data variables did not improve the model.

One patient had no available medical record (missing at random) and one did not have arthroplasty surgery (but had osteosynthesis for a hip fracture) and we assume that it was faulty registered in the SHAR. Both patients were excluded from the study. For the AE data (outcome data) we used retrospective record review (RRR), which is considered to be the most reliable method for measuring AEs. But, of course even RRR is limited by the information in the medical records.

2 Q: Several different models were tested, with rather similar outcomes on sensitivity and specificity. As you speculate in the discussion this might be caused by the limited number of variables include. Could the model(s) be improved by adding any additional and often available variables? Which variables do you expect to improve the model most?

A: There might be different ways of improving the AE measure for certain AEs. If a patient is prescribed drugs like antibiotics or high dose-anticoagulants, this could be a proxy for infection or thrombotic events. One way to extract this proxy is from the Swedish Prescribed Drug Registry that contains data on all prescribed drugs delivered to patients from Swedish pharmacies. This could be a possible addition to our model. 

3 Q: Length of stay at the orthopaedic department is an important variable in your best model, especially in the Swedish situation so you describe in your discussion. Can you elaborate on this ‘length of stay’ variable. Does it possibly contains several unmeasured factors bundles as 'length of stay' at the orthopaedic department? And therefore, how generalizable is your model to other countries?

A: This is a very important concern, and we would like to elaborate on our thoughts: The bundling of multiple factors has been one of our central hypotheses when planning this study. Length of stay is a complex factor with multiple causes and in some countries we believe that there is still a minimal stay required for reimbursement. In the vast majority of countries, length of stay is kept to an absolute minimum as hospital beds are expensive and an obvious place where money can be saved. This development has continued since the onset of the study and today's stay beyond the standard days is most likely even more associated with some kind of unmeasured factor that has been bundled into this single measure. The benefits of length of stay are (1) it is highly available in existing data collection and (2) it is difficult to manipulate - attempting to retrieve the possibly unmeasured factors is most likely only feasible as a research project, while length of stay is already available on a nationwide scale. We believe that this effect of length of stay is not specific to the Swedish setting and the results should be highly generalizable although you may need to fine-tune the model for each country. Fine-tuning is most likely also required for the Swedish setting as the mean length of stay may have changed somewhat since data collection, although fine tuning generally requires fewer data points and could thus be feasible with a much smaller data set.

4 Q: Do you have any suggestions as how to use the results of your study to improve orthopaedic care? Can you elaborate on that?

A: Using easy-accessible and reliable data on AE can improve the quality-of-care at hospitals. Swedish National Registers have practiced open accounting on hospital level for more than one decade. We believe this will initiate a competitive strive to be “best in class”, as long as participating centers actually rely on the results. 

Our model would penalize hospitals for lengthy admissions and readmissions, and we believe that the orthopedic care would be improved if both of these were reduced: 

(1) as adverse events most likely increase length of stay the indirect effect should be that hospitals increase their efforts to reduce these as well as other reasons for prolonged stay. (2) similarly, there will be an incentive for making sure that patients are optimally prepared for surgery, e.g. a diabetes patient with poor glycemic control will more likely be readmitted early on than one where the diabetes is under control.

5 Q: I am pleased with the Figure 1 showing clearly the data and analysis steps. I would suggest to use the same y-axis scale in figure 4 and 5 and clearly state in the title that it concerns Risk of adverse events associated with age/LOS. 

A: Thank you for this constructive comment, we have adjusted the mentioned figures.

6 Q: In my opinion figure 6 is not so informative and these results could be mentioned in the text of the Results section (an/or figure in an appendix).

A: We agree that this figure might be superfluous and that the time period that the data covers is too short to spot any real trends and have decided to omit the LOS trends from the paper.

Reviewer #2: 

General comments:

1 Question: It is a very technical paper, and it is hard to follow the red thread from introduction to conclusion. Many models are evaluated on the same data set, and commented on throughout the manuscript, but this study is not on the performance of different AI models, but whether administrative data captures AE better than ICD codes (to my understanding, or do I miss something?). But this also needs to reflect in the title, is it model development or adm. data vs ICD testing or perhaps both you look into.

Answer:

This is correct, the primary aim of the paper is to show that we do not have to rely on ICD-codes for measuring adverse events at a hospital level. We agree that the paper has a technical feel to it with a large set of models that can be somewhat overwhelming to readers. It is though also important that all these models perform similarly, this is what we would expect if there were a simple underlying truth and this connects to the earlier point; that we can use non-ICD administrative data to measure the adverse events. We have tried to revise the text accordingly.

2 Q: I would suggest revision of the entire manuscript with focus on a single chosen model (the best performing) and simply explain, perhaps in an appendix, that the model was chosen based on sensitivity analysis of many models, and explain the models in the appendix. And then focus the manuscript on this models performance against the gold standard of an ICD code model. Which is what I believe is the authors purpose??

A: See above answer. We agree that focusing on a single model would perhaps be beneficial but it will at the same time leave one of the central questions unanswered that most with a machine learning background will have, “does the data contain more complexity than meets the eye?”. As this paper will most likely be most of interest to an audience more heavily invested in statistics than the regular orthopedic surgeon, we would prefer to retain the current, slightly wider focus. We have revised the text accordingly and only kept the most important details about the model development and moved the full technical description to an appendix.

3 Q: The statistical modeling is complex, and beyond the abilities of this reviewer to evaluate in specifics. A notion of the experience of the author group in DL modeling would thus be nice. And also a passage on how easy the model can be constructed for use in other healthcare systems?

A: We have been working with machine learning, especially with deep learning since 2014. Prior to that MG had been developing non-linear models for modeling other complex statistical together with the Swedish Hip Arthroplasty Registry. Machine learning is a complex topic and there is a vast number of models that most clinicians will not be familiar with. The models in this paper were though rather basic since the complexity of the input data is nowhere near that of our deep learning research. E.g. a single image fed into our deep learning network for radiographs has 256 x 256 values, i.e. more than 60 000 values, while we here only had 16 values to work with. This was expected and a little simplified we evaluated with the models if there were (1) higher level interactions and (2) advanced non-linearities. As we mentioned earlier, the data followed a rather straight-forward manner without a lot of complexity and thus the statistical model that we suggest is a slightly more advance logistic regression.

4 Q: It is not clear why the individual administrative data was selected for modelling? Where they simply available or did the authors reflect a priori on this. Needs to be adressed.

A: The choice of individual administrative was made on purpose. We never considered looking using non-individual data, e.g. at averages over large groups, as we have individual data in our national registries. Furthermore, when designing the study, we were aware of the limitations when using diagnostic codes. For example, faulty coding, a multitude of possible codes for common AE such as renal failure, difficulties to separate codes given for unchanged chronic conditions and codes given for an acute deterioration and DRG creep. Therefore, we looked at which more reliable data variables that were available for all patients in easy-accessible registers and designed our models accordingly. 

5 Q: I would like if the authors state their opinion on what is next - is this model only applicable to Swedish system or should it be validated locally before application. And what is the clinical perspective of the model, is it only for use in administrative settings - such as comparison between centers, or can clinician locally or nationally benefit from this model?

A: We agree that a clinical perspective always should guide the researcher. Please see our answer to Reviewer 1 on his 3rd and 4th question. We are hoping that the concept of the model, possibly with some fine-tuning, can be implemented at a national level and allow us to evaluate in a few years if the number of adverse events drop.

6 Q: A discussion of whether the lower specificity of the new model compared to the ICD model is actually beneficial in the context of use by administrators to oversee surgeons.

A: Yes, the slightly lower specificity could be beneficial as surgeons will not know exactly what will affect the length of stay and readmissions and thus will have to apply a wider battery of improvements than just targeting single ICD-codes. We also hope that the decoupling of ICD-codes and penalties/reimbursements will improve surgeons ICD-coding and in time as this will help them to understand addressable causes for increased length of stay. 

7 Q: Do the authors plan to perform a validation of the model in a low prevalent AE group as well?

A: This is an excellent suggestion. However, the record review is very time consuming and involves a lot of administrative work that none of the authors have any plans to endure once more. We have presented the model to the Swedish Board of Health and Welfare and they might have the capacity to validate the model.

Specific comments:

8 Question: line 36: must be 30 days instead of 90 days?

Answer: Thank you for pointing this error out. It should say 30 days, we have revised the abstract accordingly.

9 Q: Line 69: I do not like the word "hypothesize" in a retrospective, exploratory study

A: We agree that this wording might be reserved for prospective studies. We have rephrased the sentence.

10 Q: Line 94-106 it is very difficult to comprehend how, and why the groups were constructed, and to what extend this influence the outcome. Why did you want to increase the probability of AE in the sample? Other causes than just for ease of journal review? 

A: This is due to the improved ability to build models when having a balanced outcome. If we would have only 5% adverse events, our models would be unable to beat the most basic model that always suggests “no adverse event”. By balancing the number of adverse events could review fewer charts and test more advanced models.

11 Q: Also later on discuss how the prevalence of AE in the selected sample influence the accuracy of your models. It is a highly selected dataset which could influence the conclusion. I do not understand Aggregated LOS data? So not individual LOS data?? You discuss later in text but it is not easily understood.

A: The aggregated data came from the Swedish board of health and welfare. It is the different percentiles of LOS divided on type of hospital, sex, fracture and year. So for each patient we added an aggregated LOS for example a 90 year old female with a fracture, treated in a university hospital in 2011 had a median LOS of x days. These variables were not used in the final model.

12 Q: line 98 add "on a dedicated sample of patients" after surgery and before the period sign.

A: We apologize, but we cannot find the word surgery in line 98, could this suggestion be for another line?

13 Q: line 112: "suffering" needs to be defined. In general this is a very loose description of AE, which makes it very difficult for others to replicate your study if desired. Needs to be more clearly stated.

A: The word suffering is included in the national patient safety terminology developed by the Swedish National Board of Health and Welfare. The Swedish adverse event definition, according to the National Board of Health and Welfare, is suffering, physical or mental harm or disease as well as death affecting a patient. If adverse events, i.e. harm, occur in a patient there are in most cases also suffering involved. If, for example, a patient is affected by a deep wound infection with a long hospital stay and reoperations there is inevitably suffering involved in connection to this along with physical harm. Suffering is closely connected to the physical harm or disease, and death part of the definition used in our study.

The adverse event definition used in this study is based on national definitions and laws (Patient Safety Act, SFS 2010:659 and Management System for Quality and Patient safety in Healthcare, SOSFS 2005:12) and the ones from many other studies which use a structured record review method. In Sweden nearly all acute care hospitals carry out Global Trigger Tool reviews on a monthly basis since 2013 (around 100 000 reviews so far) and they use nearly the same definition as we have done in this study so we believe that the study can be replicated by others.

14 Q: line 113-114. - please define "inevitable consequences". See line 112 above.

A: The word inevitable is also based on the definition in the law “Management System for Quality and Patient safety in Healthcare”, SOSFS 2005:12, but also from the one recommended by WHO in their term Healthcare-associated harm, and the one used in the Harvard Medical Practice Study (HMPS) and its subsequent nation-wide studies around the world. The HMPS record review methodology was used in the national adverse event study carried out by the Swedish National Board of Health and Welfare in 2007. It means that the harm is associated with healthcare-related omissions or commissions rather than an underlying disease, treatment or injury.

15 Q: line 115-117 why is the definition of index admission placed under AE definition subheading?

A: Thank you for pointing this out. We agree and have revised the subheading to simply; definitions.

16 Q: line 123 & line 130 I do not understand the difference in the context of this study. Why is this necessary?

A: Please see our answer concerning the aggregated data.

17 Q: Line 134 why is LOS trends valuable information in this study?

A: LOS have constantly been decreasing in Sweden and is now probably about as short it can be (or at least in some form of plateau phase). A patient that sustains an AE will most likely stay longer (this is truer for serious AEs) and because of the compressed LOS this can be used as a proxy variable for AEs. 

The idea behind the LOS trends was to highlight the decreasing LOS trend in the available data. However, since we only had available data for three years it might be to short to spot trends. We have decided to omit the LOS trends from the paper.

18 Q: Line 161-162 rephrase to a more scientific description than "we tried to"....

A: Thank you for this improving remark, we have rephrased according to your suggestion.

19 Q: Line 204-205 Does this not mean that we should use BOTH administrative data AND ICD codes in future models, instead of only either? Needs to be discussed.

A: When the ICD codes were included as a variable in the model the accuracy was lower than only the administrative variables. A plausible explanation is that the ICD code model has a very low sensitivity and the AE cases that are correctly identified by the codes are severe cases that also has a prolonged LOS and readmissions and will thus only add noise and not strengthen the model. Our recommendation is that only administrative data is used.

20 Q: Line 220 Only 2/3 were elective, non-fracture patients. Needs to address the consequences - sensitivity analysis. : line 220 "The precision was higher for all patients than for both acute and elective patients." what does this mean for your model and for external validity?

A: We conclude that this effect is due to the size of the data. The model trained on the whole training set have higher accuracy than the two models trained on the two subsets. The two cohorts apparently have more similarities than differences and therefore the models trained on 1/3 and 2/3 of the data will have lower accuracy. 

21 Q: Line 226-227: Needs to be discussed in the discussion section as to potential pitfalls of the model.

A: Please see our answer above, concerning ICD codes included in the model.

22 Q: Line 248 "We found that the risk for sustaining an AE increased with longer LOS" - I would not use the word "risk of sustaining". I would use risk for having an registered AE occur increases with longer LOS. Sustaining an AE during admission will lead to longer LOS, LOS does not increase the risk of sustaining an AE during admission in your data set. also since line 253 states: This is logical considering that most AE occurred during the orthopaedic stay.

A: Thank you for this valuable comment. We agree that it is the AE that prolongs the LOS and not the prolonged LOS that increases the risk of an AE, and this is what we tried to formulate. We have rephrased the wording according to your suggestion. 

23 Q: line 268 "the AEs were collected with RRR." Could AEs occur without being registered. I would use term "validated" instead of collected, and add sentence about missing potential AEs. 

A: Thank you for this constructive remark. We agree that RRR can only catch the AEs that are mentioned in the medical record, and although probably most AEs are mentioned we will never know how many that are not mentioned. This is mentioned later in the discussion and we have now moved that sentence to follow the passage mentioned in your question. We have rephrased according to your suggestion.

24 Q: line 262 and line 269 "An interesting finding is that the model with less variables performed better than when all variables were included. ". You place this under strength of your study, but you lack any further discussion into this. This goes as well for the complexity in the model, which gives a worse performance. These two findings has also been found by others in similar AI fields (Lauritsen et al. 2020 Artificial Intelligence in Medicine.). The manuscript would benefit with a discussion of this.

A: Thank you for highlighting this. We agree that adding a comment on this in the discussion will improve this paper. We have revised accordingly.

---

## [Decision Letter · Decision Letter 1]

16 Oct 2020

PONE-D-20-09967R1

Measuring adverse events following hip arthroplasty surgery using administrative data without relying on ICD-codes

PLOS ONE

Dear Dr. Magneli,

Thank you for submitting your manuscript to PLOS ONE. After careful consideration, we feel that it has merit but does not fully meet PLOS ONE’s publication criteria as it currently stands. Therefore, we invite you to submit a revised version of the manuscript that addresses the points raised during the review process.

We look forward to receiving your revised manuscript.

Kind regards,

Liza N. van Steenbergen

Academic Editor

PLOS ONE

Reviewers' comments:

Reviewer's Responses to Questions

**Comments to the Author**

1. If the authors have adequately addressed your comments raised in a previous round of review and you feel that this manuscript is now acceptable for publication, you may indicate that here to bypass the “Comments to the Author” section, enter your conflict of interest statement in the “Confidential to Editor” section, and submit your "Accept" recommendation.

Reviewer #1: All comments have been addressed

Reviewer #2: (No Response)

2. Is the manuscript technically sound, and do the data support the conclusions?

Reviewer #1: Yes

Reviewer #2: Yes

3. Has the statistical analysis been performed appropriately and rigorously? 

Reviewer #1: Yes

Reviewer #2: Yes

4. Have the authors made all data underlying the findings in their manuscript fully available?

Reviewer #1: Yes

Reviewer #2: Yes

5. Is the manuscript presented in an intelligible fashion and written in standard English?

Reviewer #1: Yes

Reviewer #2: Yes

6. Review Comments to the Author

Reviewer #1: The manuscript is adjusted according to my comments. I consider this manuscript ready for publication.

Reviewer #2: I agree with the authors in relation to my Q12. No further change needed in the revised manuscript.

In relation to line 121 in revised manuscript. I would ask the authors to add the definition of "suffering" to the text as defined in A to my Q13, and likewise with "inevitable consequences" as defined in A to Q14.

7. PLOS authors have the option to publish the peer review history of their article (what does this mean?). If published, this will include your full peer review and any attached files.

Reviewer #1: **Yes: **Liza van Steenbergen

Reviewer #2: **Yes: **Jeppe Lange

---

## [Author Response · Author response to Decision Letter 1]

20 Oct 2020

Question: 6. Review Comments to the Author

Reviewer #1: The manuscript is adjusted according to my comments. I consider this manuscript ready for publication.

Reviewer #2: I agree with the authors in relation to my Q12. No further change needed in the revised manuscript.

In relation to line 121 in revised manuscript. I would ask the authors to add the definition of "suffering" to the text as defined in A to my Q13, and likewise with "inevitable consequences" as defined in A to Q14.

Answer: Thank you for this constructive comment. We have added the suggested definitions.

---

## [Editor Report · Decision Letter 2]

26 Oct 2020

Measuring adverse events following hip arthroplasty surgery using administrative data without relying on ICD-codes

PONE-D-20-09967R2

Dear Dr. Magneli,

We’re pleased to inform you that your manuscript has been judged scientifically suitable for publication and will be formally accepted for publication once it meets all outstanding technical requirements.

Kind regards,

Liza N. Van Steenbergen

Guest Editor

PLOS ONE
---

## [Editor Report · Acceptance letter]

28 Oct 2020

PONE-D-20-09967R2 

Measuring adverse events following hip arthroplasty surgery using administrative data without relying on ICD-codes 

Dear Dr. Magneli:

I'm pleased to inform you that your manuscript has been deemed suitable for publication in PLOS ONE. Congratulations! Your manuscript is now with our production department. 

Kind regards, 

on behalf of

Dr. Liza N. Van Steenbergen 

Guest Editor

PLOS ONE